# Incentives and barriers to private finance for forest and landscape restoration

**Sara Löfqvist** [1,2] ✉, **Rachael D. Garrett** [2,3] ✉ **& Jaboury Ghazoul** [1] ✉

Increased private finance can accelerate forest and landscape restoration globally. Here we conduct semi-structured interviews with asset managers, corporations and restoration finance experts to examine incentives and barriers to private restoration finance. Next, we assess what type of restoration projects and regions appeal to different private funders and how current financial barriers can be overcome. We show that market incentives for corporations include meeting net-emission-reduction commitments, impact and sustainable branding opportunities, and promotion of sustainability in supply chains. Conversely, asset managers face stronger barriers to investing in restoration as it is deemed a high-risk, unknown investment with low profitability. We find that investment finance biases towards restoration projects in low-risk areas and corporate finance towards areas with business presence. Both private finance types tend to omit projects focusing on natural regeneration. Through expanded and diversified markets for restoration benefits, strong public policy support and new financial instruments, private finance for restoration can be scaled for a wider variety of restoration projects in more diverse geographical contexts.

Restoring the world's degraded forests and landscapes is imperative to safeguard ecological processes and well-being for current and future generations. Restoration is also an important nature-based solution to climate change, although uncertainties remain around the scale of emission reductions that restoration can provide[1]. In recent years, there has been a growth in national and international policy attention towards restoration. Numerous commitments and pledges have been agreed upon to catalyse restoration globally, such as the Bonn Challenge, aiming to restore 350 million hectares of land by 2030, the 1 Trillion Trees Initiative, aiming to grow, protect and restore a trillion trees by 2030, and the United Nations Decade of Ecosystem Restoration, aiming to catalyse restoration globally during the present decade. Yet only around a fifth of land pledged to be restored by 2020 had been brought under restoration as of 2019[2], and a recent progress report shows that countries are off track for meeting restoration targets set for 2030[3].

Previous restoration studies focused on ecological aspects of restoration[4–6], mapping the spatial potential for restoration[7–9] and investigating the social processes that influence and are influenced by restoration outcomes[10–13]. Some studies have also explored restoration cost and benefit structures[14,15] and financial mechanisms for restoration[16]. All of these studies provide crucial information on the potential of restoration and its possible sustainability outcomes.

There has been considerably less attention on the global pull factors needed to promote restoration. In particular, a lack of finance is one of the key barriers to upscaling restoration to meet global targets[17,18]. Most finance for restoration currently comes from public budgets[19], but these funds are too limited to support restoration needs, and they compete with a wide array of other public commitments. Since private actors have a strong influence over landscape changes through their investment decisions[20], they can potentially play a large role in complementing public-sector activities to enhance global restoration efforts.

There is a growing interest from asset managers to invest sustainably[21]. To date, 128 banks from 41 countries, holding in total US$74 trillion (around 40% of global banking assets), have committed to the industry-led and UN-supported Net-Zero Banking Alliance aiming to

[1]Ecosystem Management Group, Department of Environmental Systems Science, ETH Zürich, Zurich, Switzerland. [2]Environmental Policy Lab, ETH Zürich, Zurich, Switzerland. [3]Department of Geography and Conservation Research Institute, University of Cambridge, Cambridge, UK. ✉e-mail: sara.loefqvist@usys.ethz.ch; rg711@cam.ac.uk; jaboury.ghazoul@env.ethz.ch

align investments to carbon neutrality by 2050[22]. The Principles of Responsible Investment, outlining commitments to environmental, social and governance standards (where consistent with fiduciary duty), has over 3,000 signatories, which in total manage assets with a value of over US$103 trillion (ref. 23).

Financial instruments such as green bonds can unlock investments for environmentally sustainable assets, but these have not yet released substantial finance for restoration. Only 5% of green bonds are allocated to investments in land, compared with more established sustainable asset classes such as renewable energy (35%), sustainable buildings (30%) and sustainable transport (18%) (ref. 24).

For corporations, restoration is increasingly gaining attention as a means to address carbon emissions and meet net-zero emission reduction goals[25]. More than 4,000 corporations have committed to the Science-Based Targets, aiming to reduce emissions in alignment with the Paris Agreement[26], and carbon offsets from restoration are increasingly recognized as a means to this end. Carbon credits traded in the voluntary market exceeded a value of US$1 billion in 2021, of which Forestry and Land Use credits accounted for 61%[27].

Furthermore, it is increasingly recognized that unsustainable behaviours can lead to reputation risks, especially for companies with consumer-facing brands[28–30], and growing public concern for environmental issues makes sustainable actions beneficial for marketing purposes[16,31].

Despite this growing interest, private-actor funding for restoration remains limited[17,32]. In 2019, it was estimated that funding for biodiversity conservation globally needs to increase by on average more than US$700 billion per year, to halt and reverse land degradation, biodiversity loss and climate change[32]. Finance for agriculture was 15 times the scale of finance with forestry objectives in countries with high levels of deforestation in 2019[2], which illustrates the magnitude of financial counter pressure restoration interventions face. This lack of restoration finance stands in contrast to the growing interest from private actors in restoration. To date, no study has (to our knowledge) explored why these funding shortfalls persist.

In this article, we examine private financial actors' and restoration finance experts' perceptions of funding potential and barriers in restoration to increase understanding on why the gap between restoration finance ambition and reality persists. Our study focuses on asset managers and corporations as two financially powerful groups of private-funding actors. We choose to make the distinction between these groups as asset managers and corporations have different objectives and therefore are likely to have different approaches to restoration finance. We ask the following questions: (1) what incentives do private actors have to finance restoration? (2) what restoration project types and regions align with these incentives? (3) what barriers do private actors face when financing restoration? and (4) how can these barriers be overcome? We investigated these questions through 30 semi-structured interviews with corporations, asset managers, non-governmental organizations (NGOs), environmental consultants, a foundation and an agroforestry initiative. We used a snowball sampling approach until saturation across key themes was achieved. The data were analysed in NVivo through thematic analysis in which themes across respondents were identified inductively. A list of economic terms relevant for this section can be found in Table 1, and an overview of the actors interviewed in this study can be found in Table 2.

## Results

We will first present incentives and barriers for corporations and then for asset managers. Insights from all interview groups underlie both sections, and the specific codes behind each statement can be found in Supplementary Appendix A. Exemplary quotes underlying each statement are presented in Supplementary Appendix B. A summary of private finance flows towards restoration and the resulting benefits are visualised in Fig. 1.

## Table 1 | Economic terms

**Economic definitions used in this paper:**

*Asset manager:* Actor investing money with the aim of Return on Investment (ROI), often on the behalf of a client.

*Corporation:* A business entity engaged in selling goods or services with the aim to make financial profit.

*Non-governmental organization:* An organization working independently from the government, most commonly with a social or environmental mission.

*Private good:* A good that yields *excludable* and *rival* benefits. This means the benefits from producing the good fall primarily to the property owner (i.e. others can be excluded), and the property owner's use of that good prevents other actors from using it (i.e. there is rivalry in consumption). Agricultural products are examples of a private good.

*Public good:* A good that yields *non-excludable* and *non-rival* benefits, meaning that the wider public can enjoy the benefits regardless of who is paying for them. Clean air and healthy forests are examples of public goods.

*Green bond:* A tradeable financial asset (also known as a security) focusing on sustainable projects. A bond is a debt instrument in which the debtor owes the creditor a debt that is to be repaid at a fixed date. The debtor is also obliged to pay cyclical interest on the debt.

Overview of economic terms used in this paper.

### Market incentives for corporations to finance restoration

We identified three incentives for corporations to finance restoration. These are (1) as a means to mitigate climate change and adhere to net-emission-reduction commitments, (2) to enhance sustainability of supply chains and (3) for impact and sustainability branding. These incentives often overlap. For example, agroforestry interventions to increase sustainability in supply chains can be counted towards net-zero emission reduction commitments while simultaneously being a means towards impact and sustainability branding (Fig. 2).

**Finance for net-emission-reduction commitments.** The potential for restoration to yield net-emission reductions that could be counted towards internal climate commitments was stated as a key incentive for corporations to finance restoration either within supply chains or externally through intermediary organizations. When companies had supply chains linked to landscapes, these types of projects were often conducted within the supply chain through insetting, for example, by integrating trees into agricultural landscapes. When restoration was driven by a net-emission-reduction agenda outside of the supply chain, active restoration and tree plantations were generally favoured as such projects were perceived to be simple to quantify and communicate. All in all, the drive for emission reductions was perceived to be a crucial funding stream for restoration; "Right now carbon is the only currency we have that directly finances restoration, besides donations. So it's a huge role, it really can't be overseen. And this year has been tremendous in the uptake especially around nature-based solutions, tree planting, reforestation and so on. (...) some businesses have this year shown as much interest in it, as in the last 12,13 years combined" (EC1).

Despite indications for a substantial growth in interest, several barriers were noted when restoration is financed for carbon objectives. Notably, there is a lack of knowledge around how different restoration interventions relate to emission reductions (Fig. 2). There is also a lack of quantification systems for many associated benefits such as biodiversity and well-being benefits. This makes it difficult for corporations to capitalize on the broader array of environmental and social benefits from projects. As uncertainties remain around the benefits different types of restoration projects deliver, it is difficult for corporations to know where to channel funding. Furthermore, because some benefits cannot be properly verified, it is difficult to count them towards any existing targets. "I think the idea is, under the Science-Based Targets, that we reduce and avoid emissions and there

**Table 2 | Overview of interviewees**

| Group | Actor | Corporate role of interviewee | Continent interviewee is based in |
|---|---|---|---|
| **Asset managers** | Impact investor specifically targeting restoration and climate change (I1), Impact investors without specific focus on restoration and climate change (I2, I3), Impact investor and advisory firm focusing on climate change (I4, I5), Pension fund (I6), Corporate and investment bank (I7), Global timber investment firm (I8), Private equity firm focusing on forestry investments in Africa (I9) | Executive director and the fund manager for the climate change fund (I1), President and CEO (I2a), Vice President, Investments (I2b), Responsible investment officer (I3), Director (I4), Founding partner and joint CEO (I5), Senior portfolio manager natural resources (I6), Country head of investment management (I7), Executive chairman and founding partner (I8), Managing partner (I9) | Europe (I1, I3, I4, I6, I7, I9), North America (I2, I8), Oceania (I5) |
| **Corporations** | Multinational dairy cooperative (CP1), Multinational food and beverage companies (CP2,3,4), Chocolate and confectionary company (CP5), Multinational chemical company (CP6), Manufacturer and retailer of outdoor wear (CP7) | Development manager of sustainability (CP1), Leader agricultural procurement team (CP2), Sustainability manager (CP3), Senior climate and land use advisor (CP4), Sustainability manager (CP5), Technology and sustainability leader (CP6), Sustainability manager (CP7) | Europe (CP1, CP3, CP4, CP5, CP7), North America (CP2, CP6) |
| **NGOs** | Sustainability focused global research NGO (NGO1), Restoration focused international NGO (NGO2), Association for conservation finance experts and practitioners (NGO3), Landscape focused international NGOs (NGO4, NGO5), Landscape focused international NGO (NGO6), International sustainability NGO (NGO7), Branch within international NGO focusing on finance for sustainable landscapes (NGO8), Organization promoting rewilding of landscapes (NGO9), Organization promoting forest research and providing policy guidance (NGO10) | Senior associate focusing on restoration (NGO1), Donor relations manager (NGO2), Executive director (NGO3), Executive director and founder (NGO4), Partnerships and communications director (NGO5), Landscape coordinator and managing director (NGO6), Consultant in the international development sector (NGO7), Project lead and founder (NGO8), CEO and founder (NGO9), Director (NGO10) | Europe (NGO2, NGO4, NGO5, NGO8, NGO9, NGO10), North America (NGO1, NGO3, NGO7), Africa (NGO6) |
| **Consultancies** | Carbon finance consultancy (EC1), Organization developing mechanisms to link financial actors to conservation (EC2) | Land use fund manager (EC1), Co-founder and chief technical officier (EC2) | Europe (EC1), Asia (EC2) |
| **Foundation** | Philanthropic foundation targeting environmental and social challenges across Asia (F1) | CEO and founder (F1) | Asia (F1) |
| **Agroforestry initiative** | Agroforestry coffee project to promote Forest and Landscape Restoration (AG1) | Associated scientist (AG1) | Latin America (AG1) |

Overview of actors interviewed in this study. The numbers in parenthesis represent the code of each interviewee.

might be an option that planting trees, at least in your supply chain, can come with carbon drawdown opportunities that may in the future be counted towards your climate or your emission reduction strategies. It's not entirely clear yet how that can work. There are no accredited methodologies to do this. So it remains a bit of a grey area at the moment" (CP4).

Some actors noted the risk that strong corporate focus on carbon could crowd out ecological and social objectives of restoration. "All of the net-zero companies are driven towards the carbon side of things. And they need a return of carbon. And the other returns kind of don't have the same weight. So, everyone races to develop carbon projects, and the other ones get left behind potentially" (NGO5).

**Finance to promote sustainability of supply chain.** For corporations with supply chains linked to landscapes, such as those acting in the coffee, cocoa or dairy industry, barriers and incentives linked to restoration partly differed from when restoration was financed for net-emission reductions outside of the supply chain. For corporations with supply chains in degraded landscapes, restoration was often perceived to be a direct business opportunity to enhance landscape productivity and a means to support farmers both directly and indirectly by increasing ecosystem functionality. When restoration was executed within agricultural supply chains, it was done through agroforestry and regenerative agriculture to, for example, restore soil function (Fig. 2).

While this type of restoration sometimes provides a justifiable business case, it was noted that restoration benefits largely are public goods where financial benefits sometimes cannot be internalized or secured. Further, lack of knowledge and quantification systems for benefits resulting from such a project poses a barrier, just as when restoration is financed for net-emission reductions (Fig. 2). High upfront

costs linked to, for example, capacity building and infrastructure can thus inhibit action if the business case of such investments is unclear or not strong enough. There is high competition to identify the cheapest possible suppliers regardless of sustainability attributes. Furthermore, if a company invests in agroforestry, they often have no guarantee that farmers will continue selling to them rather than seeking other buyers.

Some interviewees referred to the hesitancy of farmers, who might not trust the new practices or worry that promised finance will not meet their expectations. Furthermore, unclear tenure makes financing of restoration risky as land could be claimed by other actors after investment. This is tied to a weak political environment and unsupportive land-use policies, often reflected by frequently changing laws and lack of transparency or enforcement of the law. Some policies, particularly subsidy schemes and policies that incentivize other land uses, may not align with restoration interests and can make farmers less willing to engage in or maintain a restoration project.

**Finance for impact and sustainability branding.** Corporations noted the value of financing restoration for its social, ecological and climate benefits alone and for associated branding benefits. With growing public environmental concern, communication of restoration can allow corporations to position themselves as being sustainable and through that gain market benefits. These motivations draw corporations towards projects with storytelling potential (to better communicate sustainability credentials), for which agroforestry or tree-planting initiatives appear especially amenable. Conversely, a lack of storytelling potential and quantification systems for benefits was a barrier to financing restoration approaches, such as natural regeneration, which did not have easily communicated pathways linking intervention to outcomes (Fig. 2).

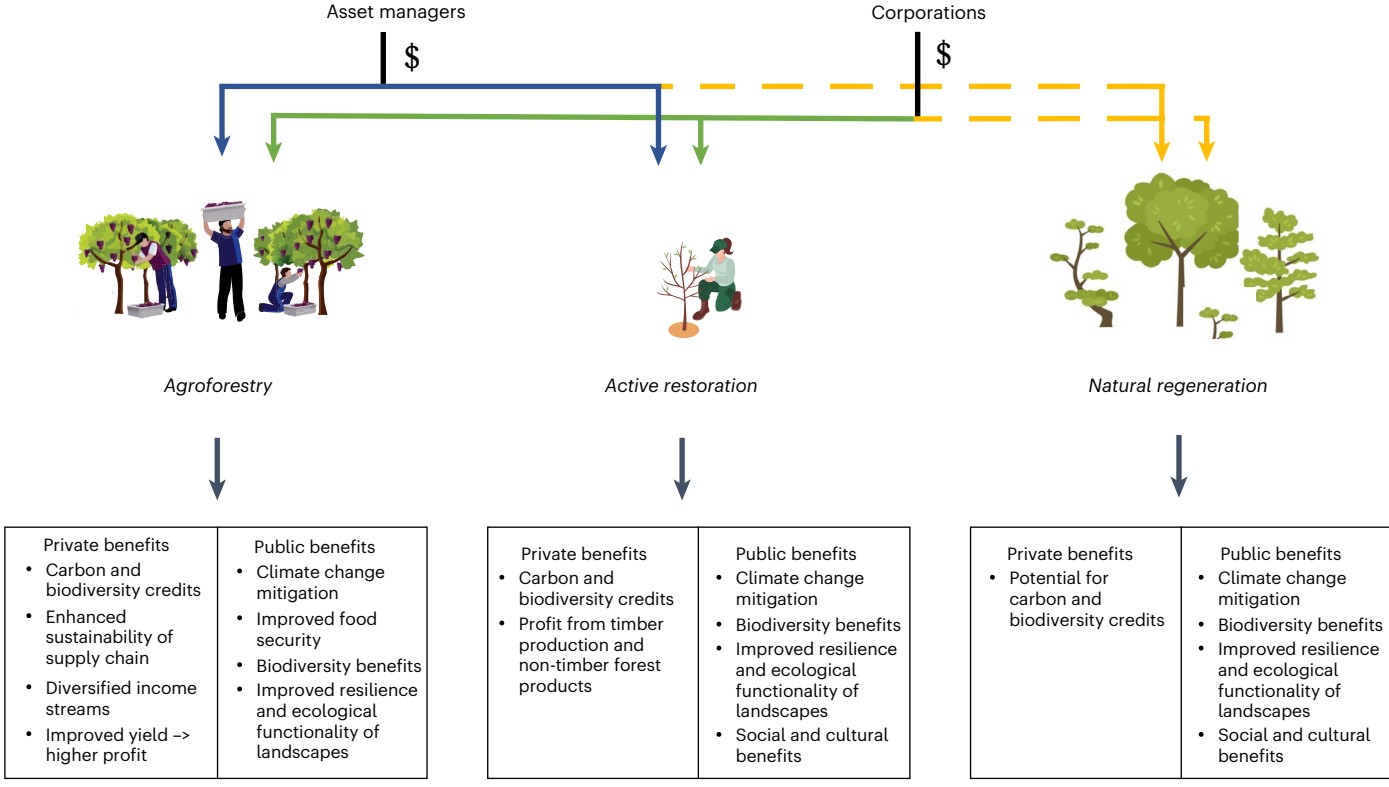

**Fig. 1 | Asset managers' and corporations' financial flows for different restoration interventions and the public and private benefits they can result in.** Green lines indicate ongoing financial flow; blue lines indicate interest but low levels of financial flow. Yellow dotted lines indicate low levels of private finance interest and financial flow, which could be ameliorated through the interventions suggested in the discussion.

## Asset managers face barriers to finance restoration

In addition to improving their sustainability profile, restoration investments can provide an avenue for asset managers to hedge risks against more unsustainable investments such as oil and gas, which can turn into stranded assets and incur reputational risks. Investments in resilience can also hedge risks from natural disasters threatening assets in landscapes. Yet none of these incentives alone will drive investment finance if the restoration project lacks a clear return on investment (ROI) profile, which is required for impact investors and conventional asset managers alike (Fig. 3).

To attract investments from asset managers, restoration projects must provide a business case with risk-adjusted ROI. This would translate into projects from which a commodity can be derived, such as timber or agricultural products, or carbon and biodiversity credits in low-risk areas.

The sustainability attribute of restoration was perceived to be a fundamental reason for why asset managers could imagine engaging in restoration. This emerges from intrinsic motivations from asset managers, as well as pressure from investors and the general public, which are increasingly concerned about the environment. "Particularly at the time when we were setting up, we were just coming out of the Paris Agreement, and it was a feeling that we all should be doing something. It was a very concrete, tangible thing to do. So, I think it's a bit of external pressure, and more genuine interest of people in these organizations. I would say it's a mix" (I1).

Risk hedging was also identified as a possible incentive for asset managers to invest in restoration. This was linked to hedging reputational risks from more-unsustainable investments to investments that are vulnerable to natural disasters and to diversifying the portfolio when being at risk of ending up with stranded assets. "For some investors and banks, where they're highly exposed to extracted markets, oil and gas or non-sustainable markets, they want to hedge that risk.

But again, from a PR perspective, but also from a pure value perspective" (I4).

Despite a budding curiosity, we found low activity by asset managers in restoration finance, and strong barriers appear to hinder this group from engaging in restoration at scale (Fig. 3). There is a mismatch with fiduciary duty, in which an asset manager is obliged to make the best financial decision for their investees. Both the financial and non-monetary benefits from restoration require a long time frame, making restoration an illiquid investment, and many projects are too small to make associated transaction costs worthwhile. The current reality is that there is a lack of bankable restoration projects that fit investment criteria, partly as the benefits of restoration are largely public goods. "There is stuff available, but it has not been de-risked enough for the likes of our fund or our big institutions to get involved in, so that it passes their internal sort of investment committee and risk criteria and solvency tests. And I mean that's the problem—these banks are not set up to invest in these sectors" (I4).

There is, in addition, a lack of standardization and knowledge around what works and what does not in terms of restoration investments. This uncertainty increases the perceived risk of restoration investments. An impact investor developing business cases for restoration stated, "We're very small and very experimental. Nobody invests in us because we have nothing. We have no track record. We don't know whether these models will work financially or not. I mean, we need to be honest, and that is the case for the majority of these restoration business models" (I1). When there is a track record, it is not always sufficient to meet investment standards. Upon discussing timber projects in African countries, a representative for a forest equity firm stated, "The return profile hasn't been there. And there has been too many failures" (I9).

Finally, restoration largely takes place in the Global South in which weak institutions, poor governance and lack of rule of law are perceived to create a difficult operating environment for any type of investment.

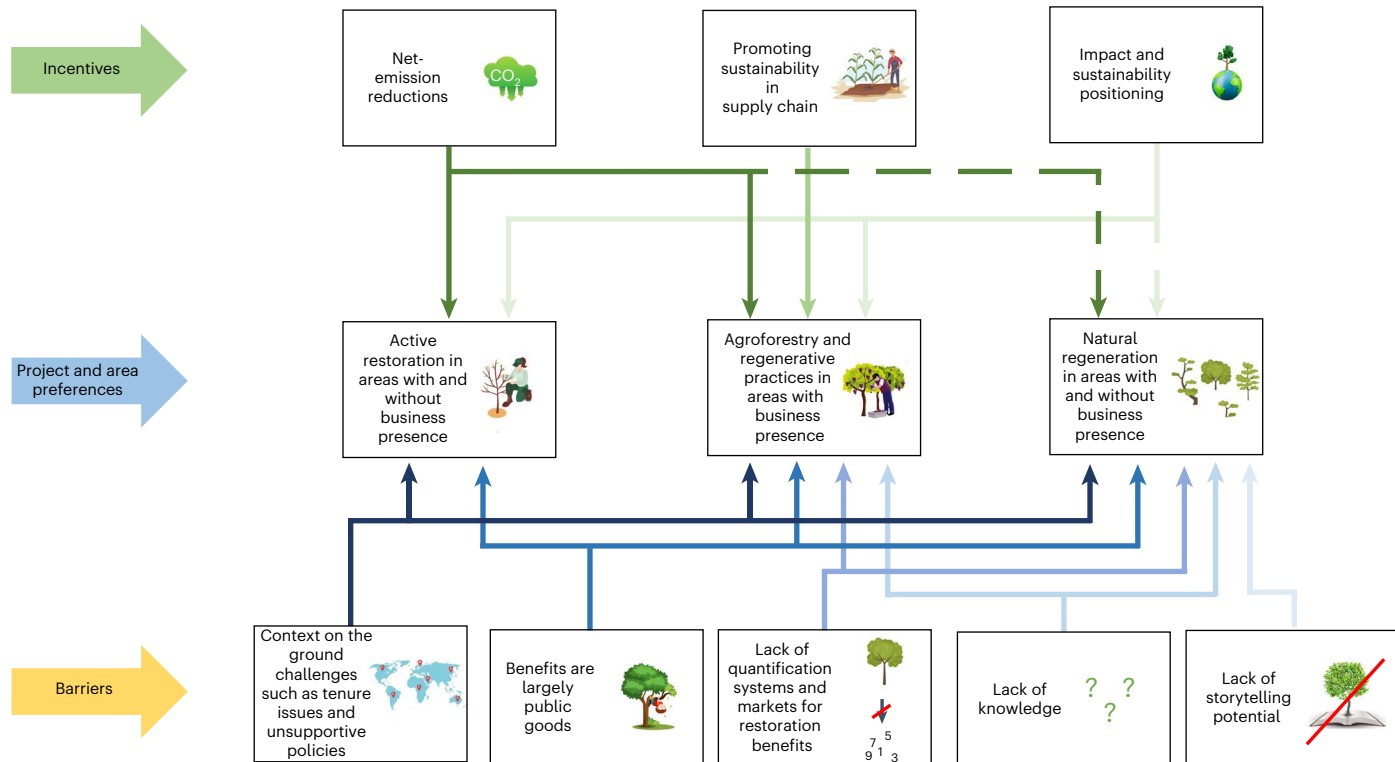

**Fig. 2 | Corporate incentives to finance restoration, explaining the types of interventions and areas these incentives match and associated barriers.** This figure illustrates the strongest incentives and barriers we found for each restoration project type and area. The arrows are in different colours for visual clarity. Dashed lines indicate incentives that are currently low but can be strengthened through interventions suggested in the discussion.

"I think, wanting to start landscape restoration in the tropics is probably a non-starter, because it's too much risk" (I6). Shifting political priorities leading to changing legislation poses another risk as it can be difficult to change project design once established. Just as for corporations, weak institutions and governance also lie behind uncertain tenure issues that represent another risk for restoration investments, as land can be claimed by other actors after an investment has been made. There is also a reputational risk that comes from not being able to ensure that investments are aligned with humanitarian rights.

## Discussion

Our study finds that market mechanisms can and do finance restoration, albeit currently not at the scale needed to meet global restoration targets. Current policy frameworks promote other types of private financial decisions and do not hold private actors sufficiently accountable for environmental harm. Thus, market mechanisms alone are unlikely to channel sufficient funding towards restoration, and better implementation of policy mandates is needed to scale funding and ensure that associated ecological and social integrity is maintained.

Corporations can have a direct business incentive to engage in restoration in production areas, through agroforestry and regenerative practices, and in non-productive areas through active restoration for net-emission reductions and impact and sustainability positioning. Yet corporations are held back by barriers such as a lack of knowledge and business case for many types of restoration projects.

Asset managers are driven primarily by ROI, unlike the broader objectives of corporations. This makes them perceive even greater barriers to financing restoration. Though there is some interest in restoration as a sustainability investment that can yield commodities, restoration is in general deemed a high-risk, unknown asset class with too-low ROI to justify those risks. Insufficient knowledge could be linked to a lack of capabilities as asset managers tend to be trained in finance and may not have knowledge about sustainability outcomes. At the same time, restoration practitioners may not have knowledge about financial realities. Few actors hold understanding of both restoration and financial markets which creates a capability void on how to scale restoration investments.

Restoration further competes with other sustainable asset classes with a proven track record and better risk-adjusted ROI profile, such as renewable energy. While both renewable energy and restoration produce positive externalities, restoration outcomes, unlike those from energy, are not always marketable.

Here we outline three public and civil society interventions that can improve the conditions for investment and corporate finance in restoration.

### Expanded markets and quantification systems for restoration benefits

Current voluntary carbon and biodiversity markets release some finance for restoration, but not at the scale needed and not always in a way that is beneficial from an ecological and social standpoint. The creation of wider restoration benefit markets together with improved systems to quantify a broader array of restoration benefits can increase private-funding incentives and steer private finance towards projects without a conventional economic business case.

Accounting for social aspects adds another challenging element to restoration benefit markets. Just as with biodiversity metrics, measuring social metrics is difficult and individual proxies can never encompass the myriad outcomes of restoration. Yet, including metrics that reflect the income and equity outcomes of restoration, as well as a wider integration of carbon and biodiversity metrics is crucial for signalling the mutual importance of these outcomes, and is likely to be important for the long-term effectiveness of restoration projects[33].

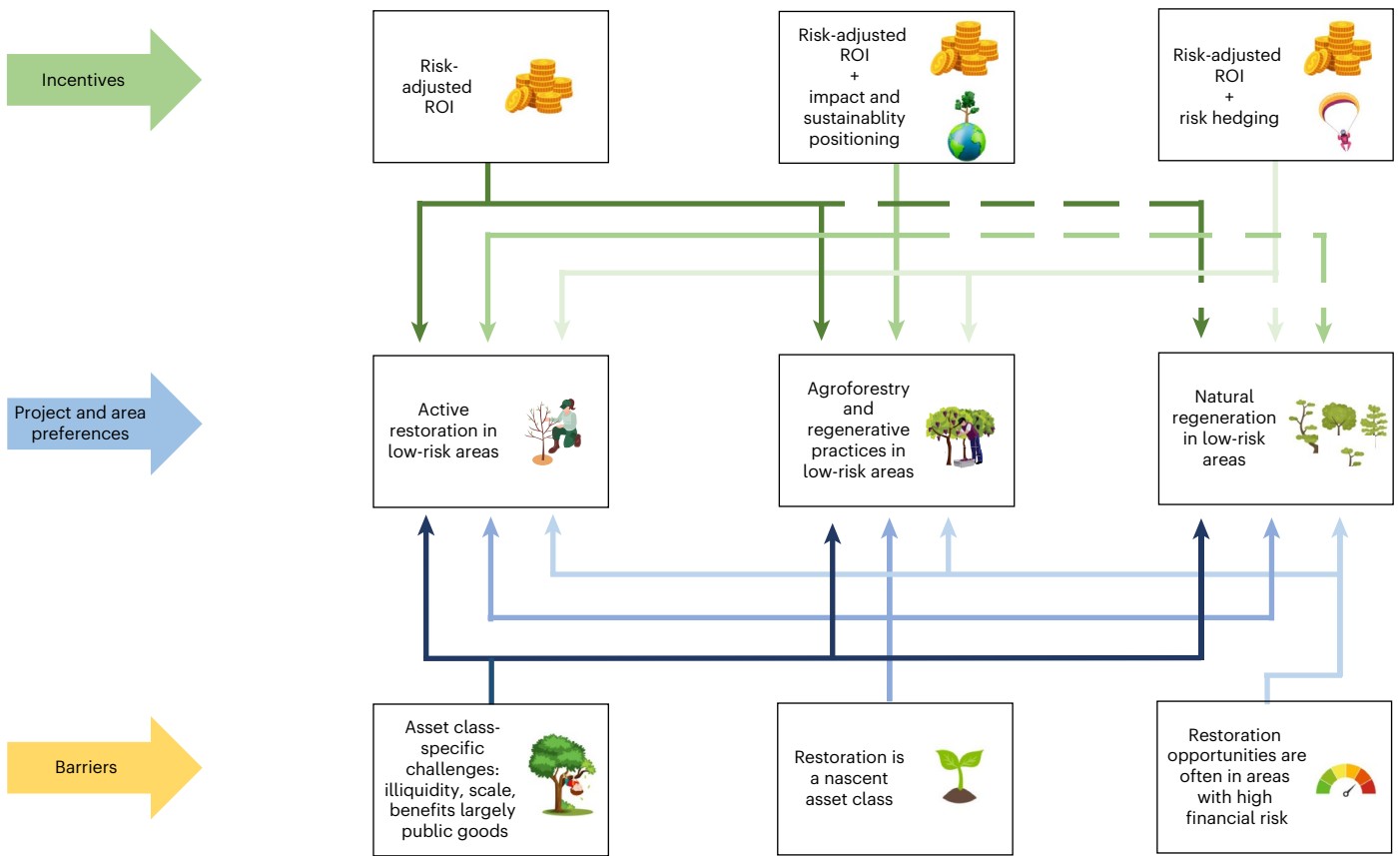

**Fig. 3 | Asset managers' incentives to finance restoration, explaining what types of interventions and areas these incentives match and what the associated barriers are.** This figure illustrates the strongest incentives and barriers we found for each restoration project type and area. The arrows are in different colours for visual clarity. Dashed lines indicate incentives that are currently low but can be strengthened through interventions suggested in the discussion.

If proper safeguards are not in place, scaling these markets introduces new risks. Markets, left to their own devices, may promote cheap, poor-quality carbon credits over more complex and expensive credits to maximize profit. Strong policy mandates are necessary to ensure that restoration credits generate ecologically sound and equitable outcomes. Policies should also emphasize net positive impact rather than offsetting, which facilitates environmental harm elsewhere. Finally, markets will be more effective if they are legally binding rather than voluntary.

## Green finance instruments and public finance

Blended finance and green bonds can spread restoration investment risk across several actors, increase the overall investment size and increase liquidity as bonds are tradable securities. One such example is the Forest Resilience Bond, a blended finance mechanism worth $25 million developed by the World Resources Institute, Blue Forest Conservation, National Forest Foundation, US Forest Service, Yuba Water Agency and the North Yuba Forest Partnership, aimed at leveraging private finance for restoration projects that promote forest resilience and post-fire restoration projects in California[34].

Public involvement in restoration provides a long-term perspective and the potential for social safeguards. Public finance can cover start-up costs linked to infrastructure and capacity building in restoration and decrease risk for asset managers by providing first-loss guarantees in blended finance schemes. Public support can also help ensure that private profit objectives work with, rather than against, the desires of the 1.4 billion people who live on areas identified to be of highest restoration priority, many of whom belong to groups with below-average levels of income, health and education[33].

There is a risk that blended finance schemes come at societal costs if public actors bear the initial costs of failed projects while private actors reap benefits or, at least, avoid losses. Thus, just as with restoration benefit markets, regulations and policy mandates are needed to stipulate private actors' investments in restoration. Blended finance mechanisms need to focus on investments that meet stringent pre-defined criteria that suggest they are financially sound and will have positive ecological and social impact, and outcomes need to be continuously monitored and verified.

Without supportive regulations, there is further risk that private funders exit after a marketable activity—often tree planting—has been completed, leaving no funding available for the maintenance that is crucial for restoration longevity. Improved monitoring and quantification systems will increase investor confidence in restoration but also hold private actors accountable to deliver on their commitments.

## Regulations and subsidies for restoration investments

Similar to the process of promoting investments in renewable energy[35], trustworthy policy signalling will be important to mobilize finance in restoration. Policy mechanisms similar to the feed-in tariffs for renewable energy[36] can provide long-term price certainty and cost guarantee for actors developing quantification systems for restoration benefit markets. For example, regulatory establishment of markets with a cap-and-trade system for biodiversity credits can increase momentum around restoration projects with stronger biodiversity profiles[37].

At the same time, private actors need better guidance on how to invest sustainably. The EU Taxonomy is a step in that direction, providing private actors and policy makers with guidance on how to channel sustainable funding, elevating protection and restoration of biodiversity and ecosystems as one of six key environmental objectives[37] The Task Force on Climate-Related Financial Disclosures is further providing recommendations for what type of information private actors should disclose in order to allow for accurate assessment of climate change-related risks and has, since its conception in 2017, seen a steady increase of companies who share such information[38].

It is important to note that neither the EU Taxonomy nor the Task Force on Climate-Related Financial Disclosures have legal mandates. Regulations that enforce sustainable behaviours, such as compulsory climate and biodiversity disclosures coupled with legally binding net-emission and net-biodiversity loss limits, have the potential to leverage faster change. At the same time, perverse governmental subsidies that enable environmentally destructive land-use practices[39] need to be redesigned to promote better social and ecological outcomes from landscapes.

Without these three strands of interventions, it is likely that private restoration finance not only will be insufficient, but will bias towards projects that focus on carbon and monoculture plantations, with uncertain or negative impacts on sustainability outcomes. As we have shown in this study, private finance is further likely to bias towards areas with business presence and areas that are deemed low risk. In this way, private finance may potentially avoid many of those areas in developing countries that have been identified to be of highest priority for restoration by previous studies[40]. This illustrates that restoration priority may not align with financial restoration feasibility, as private finance under some circumstances is likely to actively target areas with lower restoration priority but better financial scenario. Trying to make restoration projects investable under current fiscal systems may skew natural systems to fit financing criteria, rather than the other way around, jeopardizing both social and ecological outcomes of restoration. Yet, supported by sound policy frameworks, private finance holds strong potential to contribute to scaling of restoration that maintains both social and ecological integrity.

## Conclusion

Increased engagement from private funders can help to scale restoration globally. In this article, we assessed why finance remains limited despite growing private interest in restoration. Although some barriers hinder corporate finance, we find that corporations perceive existing market-driven incentives to engage in agroforestry, regenerative agriculture and active restoration to comply with emission reduction commitments, to improve the sustainability of their supply chains and for impact and to enhance their branding. Key to this is often a clear business presence and case in the target region. However, asset managers perceive mostly barriers, including the fact that restoration is a nascent, high-risk asset class, with too-low ROI to justify those risks. Asset managers favour projects in low-risk environments where there is a clear product that can be commercialized, but they note that few restoration projects fit that criteria. No actors exhibit notable interest in natural regeneration. Three strands of public intervention can help overcome these barriers: expanded markets for restoration benefits, development of green finance mechanisms and support from public finance, and regulations and subsidies for restoration investments. Through this type of public and civil society involvement, private finance can better be leveraged towards restoration that is equitable and ecologically sound.

## Methods
### Interview guide and sampling
The interview guide was developed on the basis of a literature review, attendance of relevant conferences (such as the Global Landscapes Forum Luxembourg 2019 and Innovation Forum 2019), and around 20 exploratory conversations with restoration finance experts.

In this project, we assessed funding potential stemming from two types of private-funding actors: asset managers investing with the purpose of gaining ROI and corporations as profit-driven entities producing products or services for consumption by other corporations, public actors or individuals.

To answer the research questions, we conducted 30 in-depth semi-structured interviews within the 6 categories presented in Table 2. The interviewees were sampled using existing networks, by snowball sampling and from attendee lists from relevant events (such as Global Landscapes Forum Luxembourg's finance session 2019).

### Interviews
The interviews were conducted online via Zoom, Skype or Microsoft Teams. Interviewees were contacted via e-mail, where the interview request was submitted together with a one-page explanation of the project as well as an information sheet for participants that had been approved by the ETH ethics commission. The interviews were recorded, subject to interviewees' permission to do so. If permission to record was not given, notes were taken throughout the interview. We used the same basis for the interview guide across all respondents, with slight alterations for the three groups: asset managers, corporations and restoration finance experts (including NGOs, the foundation, the agroforestry initiative and the environmental consultants). The three versions of the interview guides can be found in Supplementary Appendix C.

### Data analysis
Our interviews were transcribed using the transcription software Otter. We used the programme NVivo, which is software for analysis of text files, to code our collected data. The data were analysed using inductive methods mixed with thematic analysis[41], where key themes are identified in interview transcripts. Thematic analysis is an approach within qualitative data analysis that allows the researcher to find themes in raw qualitative data. With thematic analysis, the researcher can identify and analyse patterns within a dataset to organize and describe the data in detail[42]. To do this, the researcher will first closely study the transcribed text to look for themes that come up repeatedly. A theme is identified as something in the text that captures something of relevance in relation to the research question[42]. It could be something repeatedly mentioned by several interviewees, or something strongly emphasized by a few. This aspect of thematic analysis allows for flexibility, which is a strength of this method, but also comes with potential biases as it relies on the scientist's own judgement. We followed a six-step process[41] to analyse our data with thematic analysis: (1) becoming familiar with the data, (2) generating initial codes, (3) searching for themes, (4) reviewing themes, (5) defining and naming themes and (6) producing the report.

In Results, interview data from corporations and asset managers are presented primarily under associated headings, but sometimes the different groups spoke of experience from working with the other; thus, there are some overlaps. Data from interviews with the other interview groups underlie both sections. As there were no clear differences in perception on our key questions between the groups when discussing corporate versus asset-management finance, we do not emphasize this difference in our study.

### Potential biases
We acknowledge that our sample is subject to self-selection bias as asset managers and corporations with interest in restoration may have been more likely to agree to participate in the study. We mitigate this bias by including interviewees who work with private funders (especially the NGOs and the environmental consultants) to indirectly capture perspectives of a wider array of private-funding actors and to capture a more critical view on private-actor engagement in restoration.

Another bias relates to the possible incentive of participants to present themselves as more sustainability oriented than they actually are. We tried to address this bias by providing anonymity to interviewees and, during the course of the interview process, by bringing up potentially more sensitive topics ourselves. In this way, we would mention that a certain perspective had emerged in previous interviews and ask whether participants had any experience of this themselves. The aim of this was to lower the barriers to discussing relevant but sensitive topics. We acknowledge that perspectives from actors that have no interest in restoration finance, and are not collaborating with NGOs or consultants on other sustainability matters, were not captured by our study.

### Ethics
This study was conducted after approval from the ETH ethics commission.

### Reporting summary
Further information on research design is available in the Nature Portfolio Reporting Summary linked to this article.

## Data availability
The coded data underlying the results can be found in Supplementary Information.

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

## Acknowledgements

We thank the 30 anonymous interviewees who took time to make this study possible. R. Zamora-Cristales (WRI) and J. Lawrence (TNC) provided valuable support in the conceptualization of the study. K. Holl (UCSC), S. Schaub (Agroscope), F. Finkbeiner (ETH Zurich), A. Gee (Imperial College London) and N. Miall (University of Glasgow) reviewed earlier versions of the paper and provided insightful comments. H. Mülhaus assisted in transcribing interviews. S.L.'s work was funded by the ETH Zürich doctoral research grant (ETH-36 19-1).

## Author contributions

S.L., R.D.G. and J.G. conceptualized the paper. S.L. conducted the interviews and analysed the data. S.L. and R.D.G. developed the figures. S.L. led the writing of the paper supervised by R.D.G. and J.G.

## Funding

## Competing interests

The authors declare no competing interests.

## Additional information

**Correspondence and requests for materials** should be addressed to Sara Löfqvist, Rachael D. Garrett or Jaboury Ghazoul.

# Reporting Summary

## Statistics

For all statistical analyses, confirm that the following items are present in the figure legend, table legend, main text, or Methods section.

| n/a | Confirmed | |
|---|---|---|
| ☒ | ☐ | The exact sample size (*n*) for each experimental group/condition, given as a discrete number and unit of measurement |
| ☒ | ☐ | A statement on whether measurements were taken from distinct samples or whether the same sample was measured repeatedly |
| ☒ | ☐ | The statistical test(s) used AND whether they are one- or two-sided *Only common tests should be described solely by name; describe more complex techniques in the Methods section.* |
| ☒ | ☐ | A description of all covariates tested |
| ☒ | ☐ | A description of any assumptions or corrections, such as tests of normality and adjustment for multiple comparisons |
| ☒ | ☐ | A full description of the statistical parameters including central tendency (e.g. means) or other basic estimates (e.g. regression coefficient) AND variation (e.g. standard deviation) or associated estimates of uncertainty (e.g. confidence intervals) |
| ☒ | ☐ | For null hypothesis testing, the test statistic (e.g. *F*, *t*, *r*) with confidence intervals, effect sizes, degrees of freedom and *P* value noted *Give P values as exact values whenever suitable.* |
| ☒ | ☐ | For Bayesian analysis, information on the choice of priors and Markov chain Monte Carlo settings |
| ☒ | ☐ | For hierarchical and complex designs, identification of the appropriate level for tests and full reporting of outcomes |
| ☒ | ☐ | Estimates of effect sizes (e.g. Cohen's *d*, Pearson's *r*), indicating how they were calculated |

*Our web collection on statistics for biologists contains articles on many of the points above.*

## Software and code

Policy information about availability of computer code

| Data collection | No software was used |
|---|---|
| Data analysis | The transcription software Otter was used for transcribing interviews, and NVivo was used for data analysis. |

For manuscripts utilizing custom algorithms or software that are central to the research but not yet described in published literature, software must be made available to editors and reviewers. We strongly encourage code deposition in a community repository (e.g. GitHub). See the Nature Portfolio guidelines for submitting code & software for further information.

## Data

Policy information about availability of data

All manuscripts must include a data availability statement. This statement should provide the following information, where applicable:

- Accession codes, unique identifiers, or web links for publicly available datasets
- A description of any restrictions on data availability
- For clinical datasets or third party data, please ensure that the statement adheres to our policy

Codes representing each respondent is available in the manuscript. In the supplementary material each statement presentated in the paper is listed together with a list of codes who made that statement. Examples of quotes backing up each statement is also available in the supplementary material. All data is made available through Figshare.

# Human research participants

Policy information about studies involving human research participants and Sex and Gender in Research.

| | |
|---|---|
| Reporting on sex and gender | We did not consider gender identity or sex in this study, as we were solely looking for the participants perspectives from the point of view of their professional roles. In the assessment of restoration finance potential we believe that gender identity and sex is not a key factor influencing behavior or perspectives, so we did not ask respondents to report it. |
| Population characteristics | See answer above. As we focused on professional perspectives we did not account for population characteristic beyond those potentially affecting professional perspectives, which we identified to be area in which participant's organization is located, what type of organization they work for, and which role they have in that organization. |
| Recruitment | The interviewees were sampled using existing networks, snowball sampling, and from attendee list from relevant events (such as GLF Luxembourgs' finance session 2019). We acknowledge the self-selection bias that our population is subject to, and that asset managers and corporations with interest in restoration may have been more likely to be interested in participating in the study. The self-selection bias in our study may increase the risk that we got a too positive outlook on potential from private finance to fund restoration, as our interviewees a)were likely interested in the topic of restoration, and b) had an interest in presenting their organization as pro-sustainable solutions. See more on how we addressed this bias under "research sample" on the next page. |
| Ethics oversight | The ethics comission at ETH Zurich approved the study |

Note that full information on the approval of the study protocol must also be provided in the manuscript.

# Field-specific reporting

Please select the one below that is the best fit for your research. If you are not sure, read the appropriate sections before making your selection.

☐ Life sciences   ☒ Behavioural & social sciences   ☐ Ecological, evolutionary & environmental sciences

For a reference copy of the document with all sections, see nature.com/documents/nr-reporting-summary-flat.pdf

# Behavioural & social sciences study design

All studies must disclose on these points even when the disclosure is negative.

| | |
|---|---|
| Study description | The study is qualitative and data was collected through semi-structured interviews. The data was analyzed in the analyzing software NVivo through thematic analysis in which key themes across respondents were identified inductively. We mixed thematic analysis with inductive methods, meaning that we started out with a conceptual framing of relevant components and interaction, but then made smaller adjustments to the interview guide as the interview proceeded and new themes emerged. The aim with this method was to gain a rich and nuanced understanding of the issue we study, given the lack of prior research that has been conducted on this topic. |
| Research sample | The research sample consists of 9 representatives from asset management firms, 7 representatives from corporations, 10 representatives from NGOs, 2 environmental consultants, 1 representative from a conservation focused foundation, and 1 agroforestry initiative. These actors hold diverse expertise in the topic of restoration finance, and the diversity of interviewees gave us a well-rounded understanding of the topic of restoration finance. We acknowledge the self-selection bias that our population is subject to, and that asset managers and corporations with interest in restoration may have been more likely to be interested in participating in the study. We mitigate this bias by including interviewees who work with private funders (especially the NGOs and the environmental consultants), to indirectly capture perspectives of a wider array of private funding actors, and also to capture a more critical view on private actors engagement in restoration. Another bias is linked to the possible incentive of participants to present themselves as more sustainability oriented than they actually are. We tended to this bias by providing anonymity to interviewees, and further on in the interview process, by bringing up potentially more sensitive topics ourselves. In this way, we would mention that a certain perspective had come up in prior interviews and ask if participants had any experience of this themselves. In this way we aimed to lower the barriers to discussing topics that had been identified as relevant but that could be percieved as sensitive. Yet, we acknowledge that the actors that have no interest in restoration finance, and are not collaborating with NGOs or consultants on other sustainability matters were not captured by our study. |
| Sampling strategy | The interviewees were sampled using existing networks, snowball sampling, and from attendee list from relevant events (such as GLF Luxembourgs' finance session 2019). We did not determine sample size at the onset of the project, but instead continued sampling interviewees and conducting interviews until saturation across key themes was reached. That is, we ended data collection when new interviews did not provide insights that had not been captured by prior interviews. |
| Data collection | Data was collected through semi-structured interviews. The interviews were conducted over zoom, Skype, or Microsoft teams, and recorded with the zoom recording and/or with the iPhone voice recorder. Notes were taken throughout the interviews. In all but one interview, the researcher conducted the interview alone, and in one interview a supervising researcher participated. The researcher conducting the interviews had been involved in designing the study and was aware of the research design and questions when |

conducting the interviews. Prior to the interview the participants received a one pager explaining the project, together with an information sheet from the IRB.

| Timing | The data was collected between March 2020 and December 2021 |
|---|---|
| Data exclusions | Two interviews were excluded from the study. Both those interviews were with impact investors. One was excluded because the outcome of the interview did not fit the research questions, instead another person from the same firm was later interviewed. The second interview was excluded because the transcription was lost. |
| Non-participation | No participants dropped out after agreeing to the study. Approximately 25% of respondents that were contacted agreed to be interviewed, the others did not respond to interview requests. |
| Randomization | Participants were not allocated into experimental groups. |

# Reporting for specific materials, systems and methods

We require information from authors about some types of materials, experimental systems and methods used in many studies. Here, indicate whether each material, system or method listed is relevant to your study. If you are not sure if a list item applies to your research, read the appropriate section before selecting a response.

## Materials & experimental systems

| n/a | Involved in the study |
|---|---|
| ☒ ☐ | Antibodies |
| ☒ ☐ | Eukaryotic cell lines |
| ☒ ☐ | Palaeontology and archaeology |
| ☒ ☐ | Animals and other organisms |
| ☒ ☐ | Clinical data |
| ☒ ☐ | Dual use research of concern |

## Methods

| n/a | Involved in the study |
|---|---|
| ☒ ☐ | ChIP-seq |
| ☒ ☐ | Flow cytometry |
| ☒ ☐ | MRI-based neuroimaging |

