## [Peer Review File · Nature Ecology & Evolution]

Peer Review Information

Journal: Nature Ecology & Evolution

Manuscript Title: Incentives and barriers to private finance for forest and landscape restoration

Corresponding author name(s): Sara Löfqvist

Editorial Notes:

Reviewer Comments & Decisions:

Decision Letter, initial version:

6th September 2022

Dear Ms Löfqvist,

Your manuscript entitled "Unlocking the potential of private finance for forest and landscape restoration" has now been seen by three reviewers, whose comments are attached. The reviewers have raised a number of concerns which will need to be addressed before we can offer publication in Nature Ecology & Evolution. We will therefore need to see your responses to the criticisms raised and to some editorial concerns, along with a revised manuscript, before we can reach a final decision regarding publication.

We therefore invite you to revise your manuscript taking into account all reviewer and editor comments. Please highlight all changes in the manuscript text file.

* If you have not done so already please begin to revise your manuscript so that it conforms to our Article format instructions at <http://www.nature.com/natecolevol/info/final-submission>. Refer also to any guidelines provided in this letter.

2[REDACTED]

Nature Ecology & Evolution is committed to improving transparency in authorship. As part of our efforts in this direction, we are now requesting that all authors identified as 'corresponding author' on published papers create and link their Open Researcher and Contributor Identifier (ORCID) with their account on the Manuscript Tracking System (MTS), prior to acceptance. ORCID helps the scientific community achieve unambiguous attribution of all scholarly contributions. You can create and link your ORCID from the home page of the MTS by clicking on 'Modify my Springer Nature account'. For more information please visit www.springernature.com/orcid.

[REDACTED]

Reviewer expertise:

Reviewer #1: social science methods, conservation

Reviewer #2: private finance and conservation

Reviewer #3: environmental economics

Reviewers' comments:

Reviewer #1 (Remarks to the Author):

This paper reports the findings of a qualitative interview-based survey exploring perceived incentives and barriers related to private finance in forest and landscape restoration. The methods are

2appropriate and adequately described with one minor exception (see below), and the sample size is sufficient for a qualitative study. For example, I would expect very different responses from members of corporate sustainability teams compared to CEOs. The broad findings coincide with the existing literature on this topic and are not in themselves innovative, but I would encourage the authors to develop some specific points in more depth that may well provide original and innovative insights.

The broad findings coincide with the existing literature on this topic in that they point to a mix of financial and non-financial motivations for private financing; to the value placed on quantification and stories for corporate reputability, and to four principal barriers: high risk, low financial returns, a lack of knowledge and weak institutions and governance in emerging markets. Areas that may provide more innovative insights with deeper analysis include (i) the differences in perspectives between the three principal types of respondents and the implications of these; (ii) the relationship between direct financial incentives and deeper intrinsic motivations (which are often personal rather than corporate); and (iii) perceived incentives and barriers related to restoration in production areas versus restoration in offset areas.

The material in the results needs some restructuring. Initially I took the two sections to be reporting on incentives and barriers respectively, but if this is the case, various chunks of the text are in the wrong section. An alternative would be to present the results for each respondent group in return, and the second section appears to do this, focusing on the perspectives of asset managers, whereas the first section includes a mixture of respondent types. There is almost no material on the perspectives of practitioners.

The categories of incentives in the first section are overlapping (the first two could also be presented as subcategories of three) and they include both direct financial incentives and deeper intrinsic motivations (mitigating climate change). It would be useful to explore the relationship between these two levels of motivations more deeply. I am not sure why ROI isn't including in this section, even though it comes through as the strongest incentive in the second section.

The only added detail that I think is needed to the description of methods is to state what kinds of staff were interviewed from corporations. This may be relevant to the difference in findings between corporates and asset managers in the incentives they mentioned.

I assume the same interview guide was used for the different types of respondents but it would be useful to state this explicitly.

The section on public policies and green finance focuses almost entirely on the need for financial support and incentives, which is difficult to reconcile with earlier statements about the mix of financial and non-financial incentives. There are also some statements that need to be more nuanced. For example: while it is true that "restoration often competes with food and income production...", agroforestry often aims to combine the two (eg see <https://doi.org/10.1016/j.oneear.2019.10.017>).

The conclusion that new, expanded markets need to be developed needs to be discussed in relation to the fact carbon markets and similar trading schemes do not provide a return other than as a result of rising share prices, and therefore it is debatable whether they can be expanded.

Finally, a more detailed statement of ethical practice is needed, including the full name of the ethical review board that approved the study and what measures were put in place in relation to consent, confidentiality and data protection. I see that there is a note that this will be provided after the peer review process.

Reviewer #2 (Remarks to the Author):

Dear authors,

This paper, on the attitudes & motivations of corporations & asset managers to restore habitat, was a great read! It's exciting to see more literature emerge in this finance-nature space.

My comments on the manuscript are as follows:

- Nice clear abstract
- No "s" at the end of "Principles for Responsible Investment"
- Perhaps it would be useful to move your definition/description of what a 'corporation' is to up here, when you first use the term, for those not familiar with the private sector?
- Paragraph 1 on page 3: "recognised" :)
- The point about financial counter pressure from ag is a great point that highlights the need for *both* the increase of funding for restoration and the need to decrease funding for unsustainable farming practices (page 3).
- In the last paragraph on page 3, could you add a sentence about why you chose to make the distinction between corporates & asset managers? Do you think they are driven by different motivators, are the organizations structured significantly differently, etc. I think you have done it because they have different purposes (ROI & production goods/services, as you have stated), but perhaps a short sentence or clause explicitly stating that would be helpful to really guide the reader through your choices & why you think it was important to capture the perspectives of both
- In Table 1, could you write out in full what "FLR" stands for in the "Actor" column of the "Agroforestry initiative"? I don't think it's defined elsewhere above this table.
- I'd add a colon before the quote at the end of page four.
- Very interesting point at the end of page five about the potential for farmers to sell elsewhere after a company invests in agroforestry. I haven't seen that come up before in this space & it's an important source of uncertainty/risk.

4- You have some really great example quotes in the supplementary table - I think there's an opportunity to refer more heavily to the table/those quotes in these paragraphs here to support your points without having to add in more quotes to the main text (I'm sure you're short on space in the main text & don't want to add too much more). Perhaps it would be a case of notating the table more e.g. labelling the "Example quotes" section as Appendix B and then the "Corporate barriers to finance restoration to promote sustainability of supply chain" as iv (because it's the fourth example section) or something so that in the main body here you could say "This is tied to a weak political environment and land use policies, often reflected by frequently changing laws, and lack of transparency or enforcement of the law (Appendix B iv)." Not a necessity, but an option for drawing more attention to your example quotes throughout the manuscript.

- This is a strong first paragraph under "Asset managers face strong barriers to finance restoration". Nice!

- A bit persnickety, but in the second paragraph on page seven (& a couple other places), I would probably say "with too low an ROI" or maybe "with too-low ROI".

- I would be interested to know whether any capability barriers were acknowledged when discussing asset managers & the "lack of standardisation and knowledge"(page seven & the discussion section). Perhaps not something you could speak to if it didn't come up in interviews, but I think there's something to the idea that a "lack of knowledge of what works" is potentially driven in part by the fact that many asset managers do not seem to have a strong environmental background (& conversely, many environmental scientists do not have a strong financial background, making that interdisciplinary space a bit of a capability gap).

- Missing a "b" in "timber projects" in the 3rd paragraph of page seven.

- Great point at humanitarian rights & the trade offs & synergies between the E & the S in ESG.

- I think the discussion around the three interventions needs to be expanded a bit to include some more measured statements on the nuances/potential pitfalls of each. While I appreciate that you want to make positive suggestions for change (& that this section is not the meatiest bit of the paper), I think it is important to note that each of these proposed interventions *could* be implemented in a way that helps the private sector to the detriment of nature. E.g.

Markets: The potential for cheap, poor quality carbon credits to become the norm in the market, due to a lack of scientific rigor & regulation, and for the same to persist in other environmental markets is a legitimate concern (e.g. the current debate around Australia's carbon credit scheme). You do mention this, but I think it is worth pressing the point that the markets need to have scientific backing & regulations not watered-down by those lobbying interests.

Public finance: There are huge potential benefits here & of course it makes sense, given the interviewees' responses around the benefits mostly being public good, rather than hard ROI for investors, but the risk of failure almost always lands on the public in blended finance. This assurance is what helps drive private investment, but it is relevant, I think, to caution that done poorly, the private sector can walk away with only benefits/no harm, leaving the public sector with the bill.

5

Regulations & subsidies: Perhaps the mention of removing perverse subsidies (see Dempsey, J., Martin, T. G., & Sumaila, U. R. (2020). Subsidizing extinction?. Conservation Letters, 13(1), e12705) would be relevant here. Or similarly, regulations that enforce certain sustainable behaviours (e.g. compulsory climate & biodiversity disclosures - although of course disclosure is not the same as action!) could be mentioned.

This is my main critique of the paper - I'm not suggesting you use all of the examples I've mentioned, but I do strongly feel that this section needs to take a more nuanced, expanded view to ensure that the suggestions you are making are firmly worded & emphasise to the reader that there need to be strict science-based interpretations of each intervention in order to be effective for preventing further biodiversity loss & climate change. Without this, I fear the interventions will read as slightly generic.

- In the figure caption for Figure 1 could you describe what the different colours of the arrows mean, especially since you don't use coloured arrows in Figure 2?

- It would be good to refer to the figures in-text to ground them a bit more. At the moment they just pop out of nowhere.

- Again in the final paragraph, when you discuss what corporations are after, and then what asset managers are after, it would be good to have a sentence about why these two classes of the private sector are different (as when you introduce them at the beginning). IT might be something like "However, because asset managers are driven purely by ROI, unlike the broader objectives of corporations/the broader spread of risk across corporations, they perceive mostly barriers ..."

- The last sentence of the conclusion I think really demonstrates why adding in some more detail on the interventions is needed. Private finance *does* lack accountability to public goals, but without firm regulations & scientific backing that push them to be accountable (some sticks along with the carrots), they won't be. There needs to be a bit more acknowledgement that there are potential downsides to designing everything to the private sector's needs, rather than requiring them to make transformational change to their risk appetites & business models.

- Please provide software references for Otter & NVivo.

- Nice, clear Methods, well described. The only thing I would say (if you have the space) is that you have some really neat details in the Reporting Summary about the way you accounted for bias that could be interesting for the reader. I'm not very familiar with the Reporting Summary, so if it is easily accessible & is often read alongside the main text, it's probably not worth repeating, but if not & you have the space in the word limit, some of the info in the "Research Sample" section is great.

- I have no comments on the Supplementary materials or Reporting Summary (other than the ones above).

All in all, an enjoyable & informative read. Once you've addressed the concerns around the interventions, the rest of the comments I've made are - I feel - pretty minor. I look forward to seeing this manuscript in print at some point in the future.

Reviewer #3 (Remarks to the Author):

Overall, I found this manuscript to be timely, novel, and clearly written. Addressing the following comments would improve the impact of the paper.

1. First, additional description of the actors goals, objectives, and operating constraints would be useful because the audience of this paper may not know what objectives an asset manager is pursuing and how those differ from an NGO or corporation. I don't think this requires a lot of additional detail, but just enough to help uninitiated readers understand the differences between the motivations guiding each actor's decision making.
2. Public goods should be defined and contrasted with private goods to help readers understand why the provision of public goods poses a challenge for financing restoration with private funding. While this may appear obvious to people with backgrounds in economics and finance, it may not be as clear to readers with other areas of expertise.
3. The section describing actions to reduce barriers should be expanded. I found the discussion to be too brief and lacking in detail. For example, the discussion on green bonds did not provide me with any idea of how green bonds actually work or how they overcome barriers cited earlier in the manuscript. This section also educated me on the benefits of public involvement in financing projects, but it did not explain how we engage governments to get them to provide that support or how to create markets for other goods and services generated by restoration. That is the type of information this section should provide to pull the entire manuscript together, but this section is currently falling a little bit short in this regard. Going beyond examples and providing actual guidance in this section would help the manuscript achieve a greater impact.

*****END*****

Author Rebuttal to Initial comments

Reviewers' comments:

Reviewer #1 (Remarks to the Author):

7This paper reports the findings of a qualitative interview-based survey exploring perceived incentives and barriers related to private finance in forest and landscape restoration. The methods are appropriate and adequately described with one minor exception (see below), and the sample size is sufficient for a qualitative study. For example, I would expect very different responses from members of corporate sustainability teams compared to CEOs. The broad findings coincide with the existing literature on this topic and are not in themselves innovative, but I would encourage the authors to develop some specific points in more depth that may well provide original and innovative insights.

The broad findings coincide with the existing literature on this topic in that they point to a mix of financial and non-financial motivations for private financing; to the value placed on quantification and stories for corporate reputability, and to four principal barriers: high risk, low financial returns, a lack of knowledge and weak institutions and governance in emerging markets. Areas that may provide more innovative insights with deeper analysis include (i) the differences in perspectives between the three principal types of respondents and the implications of these; (ii) the relationship between direct financial incentives and deeper intrinsic motivations (which are often personal rather than corporate); and (iii) perceived incentives and barriers related to restoration in production areas versus restoration in offset areas.

Thank you so much for taking time to review our paper, and for raising several important points here. We're listing the responses to each comment below:

(i): We did not find any clear differences in perceptions between the different interview groups, which is why we decided to not emphasize this in the paper. Interestingly, all groups had fairly similar perceptions on the topics we examined, regardless of which group they belonged to. For example, when an NGO talked about investment barriers in restoration, their perceptions very much aligned with what the asset managers themselves had to say. After reading this comment we acknowledged that this was not explicitly stated, and in the beginning of the result section we now state that the codes underlying each section can be found in the supplementary material.

(ii) Your point two is really interesting but unfortunately we do not have sufficient data to present this in the result section.

Your point (iii) is great. We discuss this implicitly in the result section for corporations, but in response to this comment we are now explicitly stating the difference between corporate incentives for restoration in production and non-production areas in the second paragraph in the discussion, starting on line 537.

The material in the results needs some restructuring. Initially I took the two sections to be reporting on incentives and barriers respectively, but if this is the case, various chunks of the text are in the wrong section. An alternative would be to present the results for each respondent group in return, and the second section appears to do this, focusing on the perspectives of asset managers, whereas the first section includes a mixture of respondent types. There is almost no material on the perspectives of practitioners.

Thanks for this important comment. The first section presents corporations, and the second is about asset managers. We realise the structure was not clear and we have now added an introductory paragraph explaining the outline of the result section where we also state that responses from all different interview groups are incorporated under each section.

The categories of incentives in the first section are overlapping (the first two could also be presented as subcategories of three) and they include both direct financial incentives and deeper intrinsic motivations (mitigating climate change). It would be useful to explore the relationship between these two levels of motivations more deeply. I am not sure why ROI isn't included in this section, even though it comes through as the strongest incentive in the second section.

This is a great comment and we have now explained that they overlap in the introduction to this paragraph, and emphasized this relationship more throughout the first section of the result section (focusing on corporations). ROI is not included in the first section as the first section focuses on corporations, and that was not identified as a key incentive for corporations.

The only added detail that I think is needed to the description of methods is to state what kinds of staff were interviewed from corporations. This may be relevant to the difference in findings between corporates and asset managers in the incentives they mentioned.

Fully agree, thank you for this great comment. We have now edited the interviewee presentation in Table 1. to also include the respective role of each interviewee.

I assume the same interview guide was used for the different types of respondents but it would be useful to state this explicitly.

We started out with an interview guide that was then slightly iterated as we progressed with the interviews and new themes emerged. There were slight differences in the interview guide between the interview guides targeting the different groups, but they were largely similar. This has now been explicitly stated in the method and we have also added the interview guides to the Supplementary material (Appendix C).

The section on public policies and green finance focuses almost entirely on the need for financial support and incentives, which is difficult to reconcile with earlier statements about the mix of financial and non-financial incentives.

Thank you for this comment. In response to this comment we have now added a paragraph in the beginning of the discussion (starting on line 530) explicitly stated why we think the financial market alone will not manage to address the multiple goals of restoration, and hence the need for public support to steer and scale finance.

There are also some statements that need to be more nuanced. For example: while it is true that “restoration often competes with food and income production...”, agroforestry often aims to combine the two (eg see <https://doi.org/10.1016/j.oneear.2019.10.017>).

Thanks for this great comment on the trade-off sentence, we agree but decided to cut this sentence all together due to the word constraint and because we felt like it deviated a bit too much from our core message.

The conclusion that new, expanded markets need to be developed needs to be discussed in relation to the fact carbon markets and similar trading schemes do not provide a return other than as a result of rising share prices, and therefore it is debatable whether they can be expanded.

Thanks for this important comment. We have rewritten this section, stating that policy mandates can create markets, and that markets backed by policy schemes will be more efficient in ensuring ecological and social integrity of credits (which a purely market driven scheme may not). And if these markets are created by policy mandates this alone will create incentives from private actors to engage (see addition starting on line 646)

Finally, a more detailed statement of ethical practice is needed, including the full name of the ethical review board that approved the study and what measures were put in place in relation to consent, confidentiality and data protection. I see that there is a note that this will be provided after the peer review process.

Yes! We will add this after the peer review process has been completed

Reviewer #2 (Remarks to the Author):

Dear authors,

This paper, on the attitudes & motivations of corporations & asset managers to restore habitat, was a great read! It's exciting to see more literature emerge in this finance-nature space.

11Thank you so much for reviewing this paper, and for this kind comment!

My comments on the manuscript are as follows:

- Nice clear abstract

Thank you!

- No "s" at the end of "Principles for Responsible Investment"

Thanks, we have fixed this

- Perhaps it would be useful to move your definition/description of what a 'corporation' is to up here, when you first use the term, for those not familiar with the private sector?

Great idea, we are now including a table (Table 2) in which we present the economic terms used in this paper, including the term corporations. This table comes in the introduction (after line 227)

- Paragraph 1 on page 3: "recognised" :)

Thanks, it's edited! 😊

- The point about financial counter pressure from ag is a great point that highlights the need for *both* the increase of funding for restoration and the need to decrease funding for unsustainable farming practices (page 3).

Thank you!

- In the last paragraph on page 3, could you add a sentence about why you chose to make the distinction between corporates & asset managers? Do you think they are driven by different motivators, are the organizations structured significantly differently, etc. I think you have done it because they have different purposes (ROI & production goods/services, as you have stated), but perhaps a short sentence or clause explicitly stating that would be helpful to really guide the reader through your choices & why you think it was important to capture the perspectives of both

Great point, as you say we made this distinction as we believed they would be driven by different motivations given that they have different business objectives. We have now stated this on line 134.

- In Table 1, could you write out in full what "FLR" stands for in the "Actor" column of the "Agroforestry initiative"? I don't think it's defined elsewhere above this table.

Thanks for spotting this! We have spelled this out in the new version of Table 1

- I'd add a colon before the quote at the end of page four.

Thanks! It's added

- Very interesting point at the end of page five about the potential for farmers to sell elsewhere after a company invests in agroforestry. I haven't seen that come up before in this space & it's an important source of uncertainty/risk.

Thanks, we agree! I also hadn't thought about it before but after hearing it explained it makes a lot of sense that this inhibits restoration investments in situations where it is difficult to make long time contracts with farmers

- You have some really great example quotes in the supplementary table - I think there's an opportunity to refer more heavily to the table/those quotes in these paragraphs here to support your points without having to add in more quotes to the main text (I'm sure you're short on space in the main text & don't want to add too much more). Perhaps it would be a case of notating the table more e.g. labelling the "Example quotes" section as Appendix B and then the "Corporate barriers to finance restoration to promote sustainability of supply chain" as iv (because it's the fourth example section) or something so that in the main body here you could say "This is tied to a weak political environment and land use policies, often reflected by frequently changing laws, and lack of transparency or enforcement of the law (Appendix B iv)." Not a necessity, but an option for drawing more attention to your example quotes throughout the manuscript.

This is a great idea. We have now edited the titles in the supplementary material to more clearly distinguish what citations belong to what statements. Instead of repeating the reference to the appendix we decided to alert the reader to this extra material in the introductory paragraph starting on line 235.

- This is a strong first paragraph under "Asset managers face strong barriers to finance restoration". Nice!

Thank you! :)

- A bit persnickety, but in the second paragraph on page seven (& a couple other places), I would probably say "with too low an ROI" or maybe "with too-low ROI".

Thanks, that seems more correct. We have edited this on all places where we use this term

- I would be interested to know whether any capability barriers were acknowledged when discussing asset managers & the "lack of standardisation and knowledge"(page seven & the discussion section). Perhaps not something you could speak to if it didn't come up in interviews, but I think there's something to the idea that a "lack of knowledge of what works" is potentially driven in part by the fact that many asset managers do not seem to have a strong environmental background (& conversely, many environmental scientists do not have a strong financial background, making that interdisciplinary space a bit of a capability gap).

This is super interesting, but it wasn't something that came up in the interviews. We still found your point very relevant and added a short mentioning of this in the discussion (starting on line 545).

- Missing a "b" in "timber projects" in the 3rd paragraph of page seven.

Thanks! Edited

- Great point at humanitarian rights & the trade offs & synergies between the E & the S in ESG.

Thank you!

- I think the discussion around the three interventions needs to be expanded a bit to include some more measured statements on the nuances/potential pitfalls of each. While I appreciate that you want to make positive suggestions for change (& that this section is not the meatiest bit of the paper), I think it is important to note that each of these proposed interventions *could* be implemented in a way that helps the private sector to the detriment of nature.

This was great feedback and we fully agree with this. We have now reworked this section with specific focus on giving it more depth and to also account for the challenges with the interventions we suggest. We also emphasized even more strongly the ecological and social risks that come from skewing nature to fit financial investment criteria (e.g. paragraph starting on line 1016).

Markets: The potential for cheap, poor quality carbon credits to become the norm in the market, due to a lack of scientific rigor & regulation, and for the same to persist in other environmental markets is a legitimate concern (e.g. the current debate around Australia's carbon credit scheme). You do mention this, but I think it is worth pressing the point that the markets need to have scientific backing & regulations not watered-down by those lobbying interests.

Thanks for this great comment. We agree this is really important and worth more emphasis. We have now expanded on this starting on line 643.

Public finance: There are huge potential benefits here & of course it makes sense, given the interviewees' responses around the benefits mostly being public good, rather than hard ROI for investors, but the risk of failure almost always lands on the public in blended finance. This assurance is what helps drive private investment, but it is relevant, I think, to caution that done poorly, the private sector can walk away with only benefits/no harm, leaving the public sector with the bill.

This is a great point and we have now expanded on the risks with public actors taking the hit (and private actors walking away scratch free) starting on line 875. Here we also discuss how this risk can be addressed by setting and adhering to pre-defined investment criteria and monitoring and verifying progress before more finance is released.

Regulations & subsidies: Perhaps the mention of removing perverse subsidies (see Dempsey, J., Martin, T. G., & Sumaila, U. R. (2020). Subsidizing extinction?. Conservation

Letters, 13(1), e12705) would be relevant here. Or similarly, regulations that enforce certain sustainable behaviours (e.g. compulsory climate & biodiversity disclosures - although of course disclosure is not the same as action!) could be mentioned.

This is a great addition and we are now discussing this starting on line 893. We including the reference you provided (that was a great read! Thanks for pointing us to it)

This is my main critique of the paper - I'm not suggesting you use all of the examples I've mentioned, but I do strongly feel that this section needs to take a more nuanced, expanded view to ensure that the suggestions you are making are firmly worded & emphasise to the reader that there need to be strict science-based interpretations of each intervention in order to be effective for preventing further biodiversity loss & climate change. Without this, I fear the interventions will read as slightly generic.

This was really helpful! We believe that this section has a lot more depth now after incorporation of your feedback.

- In the figure caption for Figure 1 could you describe what the different colours of the arrows mean, especially since you don't use coloured arrows in Figure 2?

Yes! We have now edited the figures so there is a new version in this version of the paper, and in this we describe that the different colors are for visual clarity.

- It would be good to refer to the figures in-text to ground them a bit more. At the moment they just pop out of nowhere.

Agreed, thanks for this. We have reworked the figures and instead of the initial two, we now have three. These are now referenced throughout the text.

- Again in the final paragraph, when you discuss what corporations are after, and then what asset managers are after, it would be good to have a sentence about why these two classes of the private sector are different (as when you introduce them at the beginning). IT might be something like "However, because asset managers are driven purely by ROI, unlike the broader objectives of corporations/the broader spread of risk across corporations, they perceive mostly barriers ..."

Thanks for this great comment. We used a similar phrasing to the one you suggested in the beginning of the discussion in order to clearly state the difference between these two types of actors starting on line 543.

- The last sentence of the conclusion I think really demonstrates why adding in some more detail on the interventions is needed. Private finance *does* lack accountability to public goals, but without firm regulations & scientific backing that push them to be accountable (some sticks along with the carrots), they won't be. There needs to be a bit more acknowledgement that there are potential downsides to designing everything to the private sector's needs, rather than requiring them to make transformational change to their risk appetites & business models.

Thanks, this is a great comment. We have now reworked this section to focus more strongly on the risks with private finance in restoration, exactly what those risks are, and being more precise on how regulations can help to keep private actors in line. The section that most strongly responds to this comment starts on line 1016.

- Please provide software references for Otter & NVivo.

I apologies for this, we have tried to find them but I have canceled both subscriptions since and I couldn't see which versions I used on my old invoices. I have reached out to Otter and NVivo to see which versions I likely used based on when I started subscribing, and we will add it in the next version.

- Nice, clear Methods, well described. The only thing I would say (if you have the space) is that you have some really neat details in the Reporting Summary about the way you accounted for bias that could be interesting for the reader. I'm not very familiar with the Reporting Summary, so if it is easily accessible & is often read alongside the main text, it's probably not worth repeating, but if not & you have the space in the word limit, some of the info in the "Research Sample" section is great.

Thank you! This was a great idea. We included a section of the bias in the main method section, we agree this is information important to elevate.

- I have no comments on the Supplementary materials or Reporting Summary (other than the ones above). All in all, an enjoyable & informative read. Once you've addressed the concerns around the interventions, the rest of the comments I've made are - I feel - pretty minor. I look forward to seeing this manuscript in print at some point in the future.

Thank you! We really appreciate you taking time to review this so thoroughly. Your comments were very helpful and really helped strengthen the manuscript.

Reviewer #3 (Remarks to the Author):

Overall, I found this manuscript to be timely, novel, and clearly written. Addressing the following comments would improve the impact of the paper.

1. First, additional description of the actors goals, objectives, and operating constraints would be useful because the audience of this paper may not know what objectives an asset manager is pursuing and how those differ from an NGO or corporation. I don't think this requires a lot of additional detail, but just enough to help uninitiated readers understand the differences between the motivations guiding each actor's decision making.

19Thanks for this great comment. We have now added a table in the end of the introduction (Table 2) explaining the key economic terms used in the paper (including definitions for asset managers, corporations, and NGOs)

2. Public goods should be defined and contrasted with private goods to help readers understand why the provision of public goods poses a challenge for financing restoration with private funding. While this may appear obvious to people with backgrounds in economics and finance, it may not be as clear to readers with other areas of expertise.

Great point. We also defined these terms in Table 2, and we also illustrate the difference between public and private goods in the new Figure 1.

3. The section describing actions to reduce barriers should be expanded. I found the discussion to be too brief and lacking in detail. For example, the discussion on green bonds did not provide me with any idea of how green bonds actually work or how they overcome barriers cited earlier in the manuscript. This section also educated me on the benefits of public involvement in financing projects, but it did not explain how we engage governments to get them to provide that support or how to create markets for other goods and services generated by restoration. That is the type of information this section should provide to pull the entire manuscript together, but this section is currently falling a little bit short in this regard. Going beyond examples and providing actual guidance in this section would help the manuscript achieve a greater impact.

Thanks for this important point. We agree and in revising this section we have placed specific emphasis on challenges with the measures we propose, but we are also trying to be more specific about exactly how these measures can be implemented in practice. We now explain what a green bond is in the new Table 2, and further added a sentence on exactly how a bond like the WRI one we cite can overcome barriers on line 858. We have also expanded on how a restoration benefit market would work in the new section starting on line 643, and we have expanded on what type of governmental interventions that could leverage change starting on line 891.

Decision Letter, first revision:

13th December 2022

Dear Sara,

Thank you for submitting your revised manuscript "Unlocking the potential of private finance for forest and landscape restoration" (NATECOLEVOL-220717007A). It has now been seen again by the original reviewers and their comments are below. The reviewers find that the paper has improved in revision, and therefore we'll be happy in principle to publish it in Nature Ecology & Evolution, pending minor revisions to satisfy the reviewers' final requests and to comply with our editorial and formatting guidelines.

[REDACTED]

Reviewer #1 (Remarks to the Author):

Thanks for your detailed responses to the various comments on your paper. You have answered all my comments very thoroughly and I think the paper is ready for publication now - it comes across really well and makes some very interesting points,

Reviewer #2 (Remarks to the Author):

Dear Lofqvist & colleagues,

It was lovely to get this paper back & read all the changes you've made. I think the other two reviewers made great points & I commend you on addressing them all so well.

You have, I think, substantially improved the paper with your additions and tweaks to reflect the nuances involved in the topic. I'm sure you were aware of all of these things, but it will now be clear to readers as well.

21Congratulations on an interesting paper that I think really adds something to the growing literature in this space. I look forward to reading it again when it is published.

Kind regards,
Natasha Cadenhead

Reviewer #3 (Remarks to the Author):

Overall, I found the revisions to be satisfactory. I appreciate the inclusion of the tables one and two describing the economic actors and the economic terms used in the paper. My other primary comment concerned the section describing actions to reduce barriers. While I appreciate the author's attempt to address my comment, I still feel that this section falls a bit short of its potential. While not ideal, it is far from disqualifying. I believe my main points have been addressed, I appreciate your time in addressing them, and congratulate you on making an interesting contribution to the restoration literature.

Our ref: NATECOLEVOL-220717007A

16th December 2022

Dear Dr. Löfqvist,

Thank you for your patience as we've prepared the guidelines for final submission of your Nature Ecology & Evolution manuscript, "Unlocking the potential of private finance for forest and landscape restoration" (NATECOLEVOL-220717007A). Please carefully follow the step-by-step instructions provided in the attached file, and add a response in each row of the table to indicate the changes that you have made. Please also check and comment on any additional marked-up edits we have proposed within the text. Ensuring that each point is addressed will help to ensure that your revised manuscript can be swiftly handed over to our production team.

****We would like to start working on your revised paper, with all of the requested files and forms, as soon as possible (preferably within two weeks). Please get in contact with us immediately if you anticipate it taking more than two weeks to submit these revised files.****

If you have not done so already, please alert us to any related manuscripts from your group that are

22under consideration or in press at other journals, or are being written up for submission to other journals (see: <https://www.nature.com/nature-research/editorial-policies/plagiarism#policy-on-duplicate-publication> for details).

In recognition of the time and expertise our reviewers provide to Nature Ecology & Evolution's editorial process, we would like to formally acknowledge their contribution to the external peer review of your manuscript entitled "Unlocking the potential of private finance for forest and landscape restoration". For those reviewers who give their assent, we will be publishing their names alongside the published article.

Nature Ecology & Evolution offers a Transparent Peer Review option for new original research manuscripts submitted after December 1st, 2019. As part of this initiative, we encourage our authors to support increased transparency into the peer review process by agreeing to have the reviewer comments, author rebuttal letters, and editorial decision letters published as a Supplementary item. When you submit your final files please clearly state in your cover letter whether or not you would like to participate in this initiative. Please note that failure to state your preference will result in delays in accepting your manuscript for publication.

Cover suggestions

As you prepare your final files we encourage you to consider whether you have any images or illustrations that may be appropriate for use on the cover of Nature Ecology & Evolution.

Nature Ecology & Evolution has now transitioned to a unified Rights Collection system which will allow our Author Services team to quickly and easily collect the rights and permissions required to publish your work. Approximately 10 days after your paper is formally accepted, you will receive an email in providing you with a link to complete the grant of rights. If your paper is eligible for Open Access, our Author Services team will also be in touch regarding any additional information that may be required to arrange payment for your article.

Please note that Nature Ecology & Evolution is a Transformative Journal (TJ). Authors may

23publish their research with us through the traditional subscription access route or make their paper immediately open access through payment of an article-processing charge (APC). Authors will not be required to make a final decision about access to their article until it has been accepted. [Find out more about Transformative Journals](https://www.springernature.com/gp/open-research/transformative-journals)

Authors may need to take specific actions to achieve [compliance with funder and institutional open access mandates](https://www.springernature.com/gp/open-research/funding/policy-compliance-faqs). If your research is supported by a funder that requires immediate open access (e.g. according to [Plan S principles](https://www.springernature.com/gp/open-research/plan-s-compliance)) then you should select the gold OA route, and we will direct you to the compliant route where possible. For authors selecting the subscription publication route, the journal's standard licensing terms will need to be accepted, including [self-archiving-and-license-to-publish](https://www.nature.com/nature-portfolio/editorial-policies/self-archiving-and-license-to-publish). Those licensing terms will supersede any other terms that the author or any third party may assert apply to any version of the manuscript.

[REDACTED]

[REDACTED]

Reviewer #1:

Remarks to the Author:

Thanks for your detailed responses to the various comments on your paper. You have answered all my comments very thoroughly and I think the paper is ready for publication now - it comes across really well and makes some very interesting points,

Reviewer #2:

Remarks to the Author:

Dear Lofqvist & colleagues,

It was lovely to get this paper back & read all the changes you've made. I think the other two reviewers made great points & I commend you on addressing them all so well.

You have, I think, substantially improved the paper with your additions and tweaks to reflect the nuances involved in the topic. I'm sure you were aware of all of these things, but it will now be clear to readers as well.

Congratulations on an interesting paper that I think really adds something to the growing literature in this space. I look forward to reading it again when it is published.

Kind regards,
Natasha Cadenhead

Reviewer #3:

Remarks to the Author:

Overall, I found the revisions to be satisfactory. I appreciate the inclusion of the tables one and two describing the economic actors and the economic terms used in the paper. My other primary comment concerned the section describing actions to reduce barriers. While I appreciate the author's attempt to address my comment, I still feel that this section falls a bit short of its potential. While not ideal, it is far from disqualifying. I believe my main points have been addressed, I appreciate your time in addressing them, and congratulate you on making an interesting contribution to the restoration literature.

Final Decision Letter:

20th February 2023

Dear Sara

We are pleased to inform you that your Article entitled "Incentives and barriers to private finance for forest and landscape restoration", has now been accepted for publication in Nature Ecology & Evolution.

Over the next few weeks, your paper will be copyedited to ensure that it conforms to Nature Ecology and Evolution style. Once your paper is typeset, you will receive an email with a link to choose the appropriate publishing options for your paper and our Author Services team will be in touch regarding any additional information that may be required

25After the grant of rights is completed, you will receive a link to your electronic proof via email with a request to make any corrections within 48 hours. If, when you receive your proof, you cannot meet this deadline, please inform us at rjsproduction@springernature.com immediately.

You will not receive your proofs until the publishing agreement has been received through our system

Due to the importance of these deadlines, we ask you please us know now whether you will be difficult to contact over the next month. If this is the case, we ask you provide us with the contact information (email, phone and fax) of someone who will be able to check the proofs on your behalf, and who will be available to address any last-minute problems . Once your paper has been scheduled for online publication, the Nature press office will be in touch to confirm the details.

Acceptance of your manuscript is conditional on all authors' agreement with our publication policies (see www.nature.com/authors/policies/index.html). In particular your manuscript must not be published elsewhere and there must be no announcement of the work to any media outlet until the publication date (the day on which it is uploaded onto our web site).

Please note that *Nature Ecology & Evolution* is a Transformative Journal (TJ). Authors may publish their research with us through the traditional subscription access route or make their paper immediately open access through payment of an article-processing charge (APC). Authors will not be required to make a final decision about access to their article until it has been accepted. [Find out more about Transformative Journals](https://www.springernature.com/gp/open-research/transformative-journals)

Authors may need to take specific actions to achieve [compliance with funder and institutional open access mandates](https://www.springernature.com/gp/open-research/funding/policy-compliance-faqs). If your research is supported by a funder that requires immediate open access (e.g. according to [Plan S principles](https://www.springernature.com/gp/open-research/plan-s-compliance)) then you should select the gold OA route, and we will direct you to the compliant route where possible. For authors selecting the subscription publication route, the journal's standard licensing terms will need to be accepted, including [those licensing terms will supersede any other terms that the author or any third party may assert apply to any version of the manuscript](https://www.nature.com/nature-portfolio/editorial-policies/self-archiving-and-license-to-publish).

An online order form for reprints of your paper is available at http://www.springernature.com/reprints

<https://www.nature.com/reprints/author-reprints.html>><https://www.nature.com/reprints/author-reprints.html>. All co-authors, authors' institutions and authors' funding agencies can order reprints using the form appropriate to their geographical region.

We welcome the submission of potential cover material (including a short caption of around 40 words) related to your manuscript; suggestions should be sent to Nature Ecology & Evolution as electronic files (the image should be 300 dpi at 210 x 297 mm in either TIFF or JPEG format). Please note that such pictures should be selected more for their aesthetic appeal than for their scientific content, and that colour images work better than black and white or grayscale images. Please do not try to design a cover with the Nature Ecology & Evolution logo etc., and please do not submit composites of images related to your work. I am sure you will understand that we cannot make any promise as to whether any of your suggestions might be selected for the cover of the journal.

You can generate the link yourself when you receive your article DOI by entering it here: <http://authors.springernature.com/share>.

[REDACTED]

P.S. Click on the following link if you would like to recommend Nature Ecology & Evolution to your librarian <http://www.nature.com/subscriptions/recommend.html#forms>

** Visit the Springer Nature Editorial and Publishing website at http://editorial-jobs.springernature.com?utm_source=ejp_NEcoE_email&utm_medium=ejp_NEcoE_email&utm_campaign=ejp_NEcoE>[www.springernature.com/editorial-and-publishing-jobs](http://editorial-jobs.springernature.com?utm_source=ejp_NEcoE_email&utm_medium=ejp_NEcoE_email&utm_campaign=ejp_NEcoE) for more information about our career opportunities. If you have any questions please click [here](mailto:editorial.publishing.jobs@springernature.com).**